# Self-supervised pretraining in the wild imparts image acquisition robustness to medical image transformers: an application to lung cancer segmentation

**Jue Jiang**[*1]                                        JIANGJ1@MSKCC.ORG
**Harini Veeraraghavan**[*1]                           VEERARAH@MSKCC.ORG

**Editors:** Accepted for publication at MIDL 2024

## Abstract

Self-supervised learning (SSL) is an approach to pretrain models with unlabeled datasets and extract useful feature representations such that these models can be easily fine-tuned for various downstream tasks. Self-pretraining applies SSL on curated task-specific datasets without using task-specific labels. Increasing availability of public data repositories has now made it possible to utilize diverse and large, task unrelated datasets to pretrain models in the "wild" using SSL. However, the benefit of such wild-pretraining over self-pretraining has not been studied in the context of medical image analysis. Hence, we analyzed transformers (Swin and ViT) and a convolutional neural network created using wild- and self-pretraining trained to segment lung tumors from 3D-computed tomography (CT) scans in terms of: (a) accuracy, (b) fine-tuning epoch efficiency, and (c) robustness to image acquisition differences (contrast versus non-contrast, slice thickness, and image reconstruction kernels). We also studied feature reuse using centered kernel alignment (CKA) with the Swin networks. Our analysis with two independent testing (public N = 139; internal N = 196) datasets showed that wild-pretrained Swin models significantly outperformed self-pretrained Swin for the various imaging acquisitions. Fine-tuning epoch efficiency was higher for both wild-pretrained Swin and ViT models compared to their self-pretrained counterparts. Feature reuse close to the final encoder layers was lower than in the early layers for wild-pretrained models irrespective of the pretext tasks used in SSL. Models and code are available at https://github.com/The-Veeraraghavan-Lab/CTRobust_Transformers.git.

**Keywords:** Lung tumor segmentation, self-supervised learning, wild and self-pretraining, robustness to imaging differences.

## 1. Introduction

Self-supervised learning (SSL) is an approach to extract useful feature representations from unlabeled images by minimizing a supervised objective through pretext tasks such as jigsaw puzzles(Zhu et al., 2020), contrastive losses(Taleb et al., 2020), image reconstruction(Zhou et al., 2023), and masked image prediction(Jiang et al., 2022). Hence, SSL pretraining followed by fine-tuning on modest sized labeled datasets has demonstrated capability to achieve highly accurate segmentation in both medical and natural image analysis tasks(Nguyen et al., 2023; Jiang et al., 2022; Tang et al., 2022; Yan et al., 2023; Zhou et al., 2021). Prior works have shown that sequential SSL pretraining on natural images followed by pretraining on curated medical datasets improved 2D medical image analysis accuracy(Hosseinzadeh

---

[*] Contributed equally

et al., 2021) and that transformers benefit more from SSL than convolutional neural networks (CNN) for image classification tasks (Hosseinzadeh et al., 2021).

More frequently, medical image applications use self-pretraining, an approach wherein SSL pretraining is applied on the same curated task dataset that is subsequently used for supervised fine-tuning. In contrast, SSL pretraining in the "wild" with large and diverse datasets that are uncurated and unrelated to task have shown to be an effective approach for natural image analysis(Matsoukas et al., 2022). The rationale for wild-pretraining is to leverage the imaging variations inherent in large and diverse sets of images to extract universally applicable feature representations for the downstream tasks. However, supervised pretraining using ImageNet has shown only variable feature reuse depending on the distance of the medical image domain from natural images(Raghu et al., 2019).

In this context, the benefits of wild-pretraining performed on medical images over self-pretraining with curated, task-specific datasets has not been studied. Hence, we studied the impact of wild-pretraining on a relatively large and uncurated 3D medical datasets ($>$ 10,000 CTs) vs. self-pretraining with curated, 3D CT dataset for segmenting lung tumors along with evaluation of robustness to imaging variations. We chose volumetric lung tumor segmentation because SSL pretext tasks focus on learning universal feature representations, which is likely to capture common elements like organs but not tumors. Hence, our chosen application of tumor segmentation allows us to study whether wild-pretraining from images encompassing wider variations benefits over self-pretraining on the task dataset.

Our contributions include: (a) Comparative analysis of SSL-based wild-pretraining and self-pretraining applied to three common architectures, a vision transformer (ViT), hierarchical shifted window transformer (Swin), and a Unet-based convolutional network for lung tumor segmentation, (b) analysis of robustness to CT acquisition differences due to the two SSL pretraining for the individual architectures, (c) evaluation of pretext tasks on the feature reuse with SSL. Understanding the relative merits of the SSL approach for commonly used networks could inform the development of pretrained models.

## 2. Datasets

Analyzed datasets with imaging acquisitions, and disease details are in Table 1.

**Wild-pretraining:** A total of 10,412 3D CT scans covering head to pelvis sourced from datasets provided publicly for variety of tasks including lesion detection (Xiao et al., 2023), classification (Harmon et al., 2020), and multi-organ and abdominal tumor segmentations were used without additional curation for pretraining. Retrospectively collected and anonymized institutional datasets were used as is from patients treated for lung, esophageal (Internal 1) and head and neck (Internal 2) cancers treated with radiotherapy (RT).

**Task dataset for self-pretraining and fine-tuning:** A publicly available dataset of patients with locally advanced non-small cell lung cancer (LA-NSCLC) scanned with contrast and non-contrast CTs, smooth reconstruction kernels ($\leq$ B30), 3 mm slices, and provided with tumor contours was analyzed (Aerts et al., 2015). Tumor sizes ranged with a median of 33.68 cc and interquartile range (IQR) of 8.29 cc to 90.31 cc. A random set of 316 CTs were used in self-pretraining without tumor labels.

**Testing:** Two independent datasets totalling 335 CTs, consisting of a public dataset of patients with early stage (stage I-II) NSCLC (Bakr et al., 2018) (median 7.91 cc, IQR of 3.60 cc to 28.23 cc) and institutional dataset of patients with stage (II-IV) NSCLC (median

19.54 cc, IQR of 6.64 cc to 66.87 cc) were evaluated. The LRad dataset used contrast and non-contrast CTs, a range of slice thicknesses, and a wide variety of image reconstruction kernels, which were categorized as smooth, medium, and sharp kernels for meaningful analysis of robustness. The institutional LC dataset was homogeneous in terms of CT acquisition (convolutional kernel used GE lung reconstruction and a slice thickness of 5mm). In addition, a subset of 20 patients reconstructed with both sharp (GE lung) and smooth (GE standard) kernel as well as with 2.5mm and 5mm slices were used for paired comparison of accuracy differences that controlled for tumor and patient anatomy differences.

Table 1. Datasets summary. Smooth kernels: GE "standard" and "bone", Siemens $<$ B40; Medium: GE "Bone Plus", Siemens $\geq$ B40 and $<$ B50; Sharp: GE "Lung", Siemens $\geq$ B50. NA: not available indicated when not provided for a dataset.

| Data | Location | Number | Manufacturer | Thickness | Kernel | Contrast |
|---|---|---|---|---|---|---|
| **Pretraining** | | | | | | |
| MELA 2022(Xiao et al., 2023) | Chest | 880 | NA | 1 mm | NA | contrast, non-contrast |
| AMOS 2022(Ji et al., 2022) | Chest-Abd-pelvis | 360 | NA | 5mm to 7.5 mm | NA | contrast, non-contrast |
| COVID-19(Harmon et al., 2020) | Chest | 609 | NA | 5 mm | NA | contrast, non-contrast |
| KITS ((Heller et al., 2019)) | Abdomen-pelvis | 411 | Siemens, Toshiba | 3 mm | smooth | arterial, late, non-contrast |
| Pancreas CT(Roth et al., 2015) | Chest-Abdomen | 80 | NA | 1mm to 5mm | NA | contrast, non-contrast |
| Internal 1 Radiotherapy | Chest | 5,124 | GE | 3 mm to 5 mm | smooth, sharp | contrast, non-contrast |
| Internal 2 Radiotherapy | Head and neck | 2,632 | GE | 2.5 mm to 3mm | smooth | contrast, non-contrast |
| **Fine-tuning/Self-pretraining** | | | | | | |
| TCIA NSCLC(Aerts et al., 2015) | Chest-abdomen | 350 | Siemens, CMS | 3 mm | Smooth | contrast, non-contrast |
| **Testing** | | | | | | |
| LRad(Bakr et al., 2018) | Chest | 139 | Siemens, Toshiba, GE | 0.9 mm to 5mm | smooth, medium, sharp | contrast, non-contrast |
| LC | Chest-abdomen | 196 | GE | 1.25 mm to 5mm | smooth, sharp | contrast, non-contrast |

## 3. Problem formulation and methodology

The aim of this work is to understand the benefits of using SSL-based wild-pretraining with large, diverse and uncurated medical images compared to SSL-based self-pretraining with curated, in-domain task-datasets for segmenting lung tumors from CT scans. Specifically, we analyzed under what conditions wild-pretraining improves over self-pretraining by studying three common networks with varying inductive bias and CT imaging differences. ViT (Dosovitskiy et al., 2021), which removes image specific inductive biases including locality and translational equivariance has the least inductive bias. Swin (Liu et al., 2021) adds back hierarchical scale and windowed attention to improve inductive bias. A CNN model such as non-skip Unet (nsUnet) (Zhou et al., 2023) has the highest inductive bias.

Feature reuse has been attributed to success in transfer learning applied to supervised pretraining with ImageNet(Matsoukas et al., 2022; Raghu et al., 2019). We studied how feature reuse is impacted by the SSL strategy as well as the pretext task used for SSL.

### 3.1. Network architectures and SSL

Transformer encoders connected to 3D Unet decoder shown to be effective for multi-organ(Hatamizadeh et al., 2022b; Tang et al., 2022; Jiang et al., 2021) and brain tumor(Hatamizadeh et al., 2022a) segmentation were studied. The 3D Swin encoder used a Swin-small backbone, which used a depth of [2,2,8,2] and [4,4,8,16] multi-head for each transformer depth, and a feature embedding size of 384. This setup also included a window size of $4 \times 4 \times 4$ and patch size of $2 \times 2 \times 2$. The ViT encoder comprised of 12 transformer blocks, 768 embedding features, 8 multi-head self attention blocks, and a patch size of 8. A non-skip Unet (nsUnet) was used as the CNN network due to it's higher accuracy over Unet for medical image segmentation(Zhou et al., 2023). The nsUnet had 4 downsampling and corresponding number of upsampling layers. All three networks used an input image of

$128 \times 128 \times 128$ voxels. The total number of parameters for ViT and Swin were 46,405,874 and 64,698,114, respectively. nsUnet had 17,111,499 parameters.

## 4. SSL pretext tasks

Wild- and self-pretraining was performed with an identical set of 5 pretext tasks that consisted of contrastive pretraining (Chen et al., 2020), masked image prediction (MIP), two self-distillation learning tasks called masked patch token distillation (MPD) and image token distillation (ITD), as well as a combination of MIP, MPD, and ITD tasks as used in the SMIT method (Jiang et al., 2022). We also studied the impact of sequential SSL by using wild-pretraining followed by self-pretraining using SMIT.

Contrastive pretext task minimized the cosine similarity of feature embeddings from positive pairs (augmented 3D views from same patient) while maximizing the same distance between negative pairs (augmented 3D image views from different patients). ITD and MPD tasks are implemented through self-distillation performed between a student and an exponentially moving average teacher network(Jiang et al., 2022), whereby, the two networks were presented with different 3D augmented views of input images. The view provided to the student was randomly masked using a default masking ratio of 0.75 (Jiang et al., 2022). MIP forces the student to correctly predict the image regions underlying the masked image patches by utilizing the visual context from the visible image patches. MIP was implemented by minimizing the L1-norm between predicted and unmasked image. MPD and ITD were implemented using cross-entropy losses with temperature scaling to match the patch feature token and image token distributions, respectively.

### 4.1. Implementation details

All images intensity clipped [-500 HU to 500 HU], normalized to [0,1] and resampled to a uniform voxel size of 2mm $\times$ 2mm $\times$ 2mm and then randomly cropped to $128 \times 128 \times 128$ voxels to generate the 3D views. Images were resampled to a uniform voxel size of 1.5 mm $\times$ 1.5 mm $\times$ 2.0 mm voxels for fine-tuning and testing. The networks were optimized using ADAMw (Loshchilov and Hutter, 2017) with a cosine learning rate scheduling (Ilya and Frank, 2016), and trained for 500 epochs with an initial learning rate of $8e^{-4}$ and warmup for 50 epochs. Self-pretraining mitigated the issue of learning from fewer examples compared to wild-pretraining by using online data augmentation and training for 2,000 epochs with a warmup for 200 epochs. A path drop rate of 0.1 was applied to the student model, and all SSL tasks were conducted on 4 NVIDIA A100 GPUs ($4\times$ 80GB memory) using a batch size of 32 for Swin, 32 for nsUnet, and 8 for ViT.

Fine-tuning was performed on NVIDIA $4\times$A100 GPU. All analyzed networks were trained with a learning rate of $2e^{-4}$ for 1,000 epochs. Swin and nsUnet models were fine-tuned with a batch size of 24 and the ViT models used a batch size of 4 due to memory limitations. Early stopping was used to select the model with highest accuracy on validation set.

### 4.2. Experiments and evaluation metrics

Tumor segmentations were compared against manual delineations using the Dice similarity coefficient (DSC). Fine-tuning epoch efficiency was measured as the relative difference in the number of epochs at which fine-tuning was stopped with respect to the number of epochs required for training the model from scratch and expressed as a percentage. Statistical

comparisons measured tumor segmentation accuracy differences between fine-tuned models produced with wild-pretrained and self-pretrained models for individual networks (e.g. ViT wild-pretrained vs. ViT self-pretrained; Swin wild-pretrained vs. Swin self-pretrained) using paired, two-sided, Wilcoxon signed rank tests at 95% significance level. Feature reuse from wild-pretrained to fine-tuned as well as self-pretrained to fine-tuned models were measured using centered kernel alignment (CKA) as detailed in Supplementary section A.

## 5. Results

### 5.1. Segmentation accuracy

As shown in Table. 2, wild-pretrained ViT and Swin models were more accurate their self-pretrained counterparts on both datasets. Wild- and self-pretrained CNN models were similarly accurate. Further analysis showed that wild-pretraining reduced dependency of tumor segmentation accuracy to volume compared to other training strategies for all three networks. Transformer methods showed smaller dependency of accuracy to tumor volume (Swin $R^2$ ranged from 0.11 for wild-pretrained to 0.14 for scratch trained; ViT $R^2$ ranged from 0.11 for wild-pretrained to 0.18 for scratch trained) when compared to CNN ($R^2$ of 0.37 for wild-pretraining to 0.41 for scratch training) as shown in Supplementary Figure. A.1. Example segmentations produced by the wild- and self-pretrained Swin following fine-tuning are shown in Figure. 1, which clearly show better performance of the wild-pretrained model. Individual pretext tasks did not lead to large differences in accuracy. However, a combination of pretext tasks as done in SMIT showed a larger accuracy improvement. Two-stage pretraining did not improve accuracy compared to wild-pretraining.

Table 2. Tumor segmentation accuracy with pretraining methods and transformer architectures.

| Model | Training | Pretext Task | LRad | LC |
|-------|----------|--------------|------|-----|
| CNN | Scratch | N/A | 0.42±0.34 | 0.54±0.24 |
| CNN | Self-pretraining | PRCLv2(Zhou et al., 2023) | 0.45±0.33 | 0.56±0.26 |
| CNN | Wild-pretraining | PRCLv2 | 0.46±0.33 | 0.57±0.20 |
| ViT | Scratch | N/A | 0.55±0.31 | 0.64±0.26 |
| ViT | Self-pretraining | SMIT | 0.64±0.25 | 0.67±0.22 |
| ViT | Wild-pretraining | SMIT | **0.66±0.22** | **0.70±0.23** |
| Swin | Scratch | N/A | 0.54±0.31 | 0.68 ± 0.24 |
| Swin | Self-pretraining | SMIT | 0.63±0.23 | 0.71 ± 0.21 |
| Swin | Wild-pretraining | SMIT | **0.69±0.18** | **0.72 ± 0.20** |
| Swin | Wild and Self-pretraining | SMIT | 0.65±0.21 | 0.71±0.21 |
| Swin | Wild-pretraining | MIP | 0.64±0.25 | 0.69±0.24 |
| Swin | Wild-pretraining | ITD | 0.64±0.24 | 0.70±0.21 |
| Swin | Wild-pretraining | ITD & MPD | 0.66±0.21 | 0.71±0.21 |
| Swin | Wild-pretraining | Contrastive | 0.64±0.24 | 0.69±0.24 |

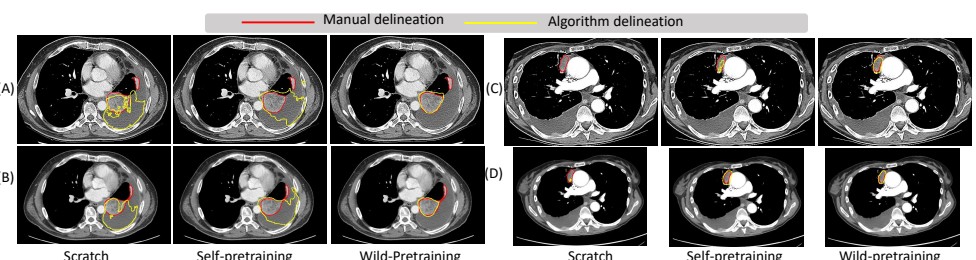

Figure 1. Segmentation (yellow contour) produced by Swin model applied to CTs reconstructed with sharp (A, C) and smooth (B, D) for two different patients with 2.5 mm thickness.

## 5.2. Robustness to CT imaging variations

Analysis of accuracy robustness to CT contrast differences showed that all three network architectures resulted in higher accuracy for contrast-enhanced CTs when using wild-pretraining compared to self-pretraining (Table. 3). Wild-pretrained Swin was significantly more accurate than self-pretrained Swin for both contrast and non-contrast CTs. There was no difference in accuracy for ViT or CNN models using the two SSL strategies.
Figure. 2 shows the impact of CT reconstruction kernels on tumor segmentation accuracy using the public LRad dataset (n=139 CTs). The wild-pretrained Swin was significantly more accurate than self-pretrained Swin (p < 0.001) with smooth kernel and scratch trained Swin with smooth (p < 0.001) and sharp kernels (p = 0.006).

Table 3. Tumor segmentation accuracy differences due to CT contrast. Significance comparisons of models were performed with respect to pretrained models on the public LRad dataset.

| Model | training | Contrast (N=85) | p-value | NonContrast (N=54) | p-value |
|---|---|---|---|---|---|
| CNN | Scratch | 0.47±0.33 | 0.014 | 0.35±0.33 | 0.28 |
| CNN | Self-pretraining | 0.51±0.32 | 0.72 | 0.36±0.32 | 0.43 |
| CNN | Wild-pretraining | **0.52±0.32** | - | **0.38±0.33** | - |
| ViT | Scratch | 0.60±0.30 | 0.00004 | 0.47±0.31 | 9.13e-6 |
| ViT | Self-pretraining | 0.65±0.27 | 0.064 | **0.65±0.19** | 0.61 |
| ViT | Wild-pretraining | **0.67±0.23** | - | 0.64±0.20 | - |
| Swin | Scratch | 0.58±0.30 | 3.26e-6 | 0.48±0.31 | 5.21e-7 |
| Swin | Self-pretraining | 0.66±0.23 | 0.0041 | 0.60±0.23 | 6e-4 |
| Swin | Wild-pretraining | **0.70±0.19** | - | **0.68±0.16** | - |

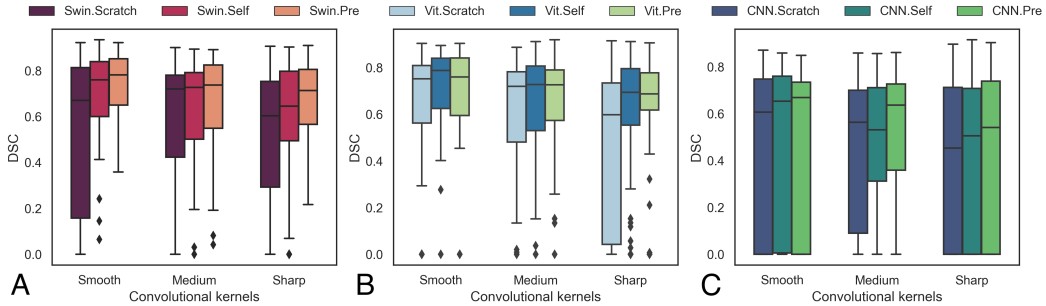

Figure 2. Influence of CT reconstruction kernel on segmentation accuracy with (A) Swin-backbone, (B) ViT-backbone, and (C) CNN backbone. Analysis was performed on all public LRad test cases. Abbreviations 'Self' refers to self-pretraining and 'Pre' refers to wild-pretraining.

## 5.3. Feature reuse analysis

There was a considerable variation in the reuse of the features for the same network architecture (Swin) when wild-pretrained with different pretext tasks as shown in Figure. 3 (A) to (E). Concretely, contrastive task resulted in the highest feature reuse even across different feature layers (off-diagnoal entries in the CKA matrix). ITD, a global image feature matching loss, resulted in the lowest feature reuse, followed by MPD, MIP, and SMIT (Figure. 3). SMIT, which uses a combination of ITD, MIP and MPD, the latter two are spatial locality context losses, resulted in higher feature reuse in the lower (1 to 4) and middle level features (5 to 9) compared to higher level (10 to 14) layer features. In addition, the features across different layers (off-diagonal entries of the CKA matrix) were different between wild-pretrained and fine-tuned features for SMIT, MPD, and ITD tasks but not for the contrastive learning task. Self-pretraining (Figure. 3 F) with SMIT resulted in more

differentiation of off-diagnoal features (layers 5 to 14 compared to lower level features 1 to 4) but such differentiation was to a lesser degree than wild-pretrained SMIT. In general, wild-pretraining resulted in feature changes especially close to the later stage encoder layers (13 and 14) for all the pretext tasks when compared to self-pretraining.

The trend of higher feature reuse for self-pretraining compared to wild-pretrained models was also observed for contrast and non-contrast CT scans (Figure. 4). In particular, wild-pretraining resulted in larger deviations of features close to the later encoder layer (13 and 14) for contrast compared to non-contrast CTs (Figure. 4 A and B). Analysis of smooth and sharp reconstruction kernels showed higher feature reuse with wild-pretrained models for the lower levels compared to later layer (13 and 14) as shown in Figure. 4 C and D.

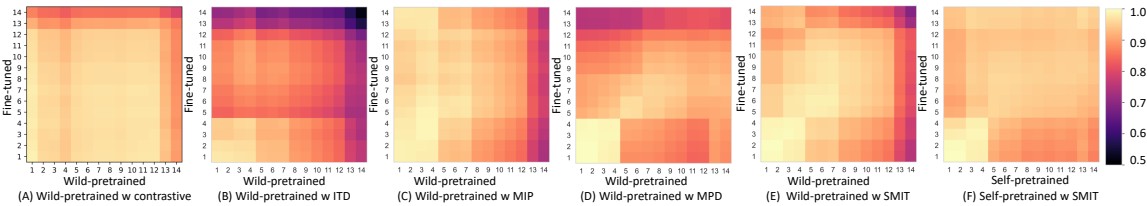

Figure 3. CKA analysis performed on the Swin network using pretraining with SMIT (D) and self-pretraining with SMIT (E) as well as pretraining with different pretext tasks including (A) contrastive (B) ITD, (C) MIP, and (D) ITD and MPD.

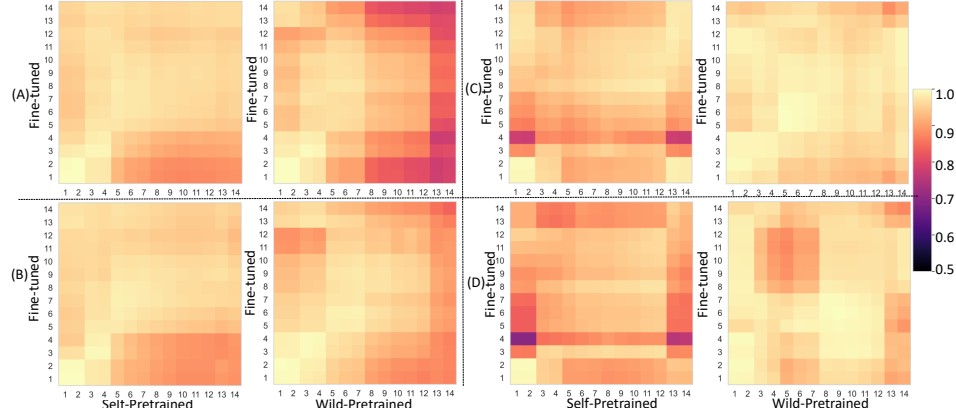

Figure 4. CKA analysis to measure the similarity features for (A) contrast and (B) non-contrast CT as well as CT images reconstructed using (C) smooth and (D) sharp kernel with 2.5 mm slices.

### 5.4. Fine-tuning epoch efficiency

Wild-pretrained models required fewer GPU hours for fine-tuning the models compared to self-pretrained models for Swin, ViT, and CNN networks (Table 4). Wild-pretrained Swin models were most efficient in terms of the number of epochs required for fine-tuning. Validation curves for the various architectures shows faster convergence of wild-pretrained models compared to self-pretrained counterparts (Supplementary Figure A.2).

### 5.5. Why wild-pretraining works better than self-pretraining?

Analysis of feature reuse between the wild-pretrained and fine-tuned features as well as self-pretrained and fine-tuned features showed that lower level features (1 to 4) were similar

Table 4. Fine-tuning epoch efficiency of self-pretrained and pre-trained models with validation DSC.

| Model | Training | DSC | Epoch efficiency % | GPU hours |
|-------|----------|-----|--------------------|-----------|
| CNN | Scratch | 0.69 | - | 79 |
| CNN | Self-pretraining | 0.70 | 30 % | 55 |
| CNN | Wild-pretraining | 0.72 | 50 % | 39 |
| ViT | Scratch | 0.67 | - | 153 |
| ViT | Self-pretraining | 0.72 | 15 % | 130 |
| ViT | Wild-pretraining | **0.74** | 47 % | 81 |
| Swin | Scratch | 0.68 | - | 130 |
| Swin | Self-pretraining | 0.73 | 46 % | 70 |
| Swin | Wild-pretraining | **0.78** | 80 % | 26 |

for both SSL approaches Figure. 3 (E) and (F). Features in the middle layers (5 to 7) were also similar for wild-pretrained model, indicating greater feature reuse for low and some mid-level layers. Also, larger differentiation of the features at the deeper layers (13 and 14) occurred for the wild-pretrained model compared to self-pretrained approach, indicating greater adaption of the network's features to the segmentation task. Feature analysis (Figure. 5 (C)) shows wild-pretraining and self-pretraining produces different pretrained features. Feature self-similarity analysis (Figure. 5 (A) and (B)) shows that high self-similarity of same features but differentiation of different layer features with wild-pretraining, indicating ability to extract a wider range of pretrained features.

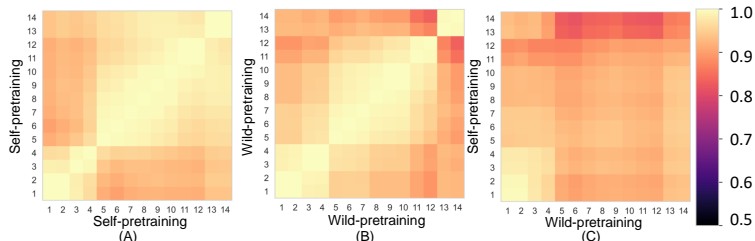

Figure 5. CKA analysis measuring feature self-similarity for self- and wild-pretrained Swin.

## 6. Discussion and conclusions

We performed a comprehensive analysis of SSL wild-pretraining and self-pretraining applied to two transformer and one CNN model in terms of accuracy, robustness to imaging differences, as well as feature reuse. Our results are consistent with findings from natural images that demonstrated improvements in accuracy and fine-tuning epoch efficiency with wild-pretraining (Goyal et al., 2021). Our analysis also showed that wild-pretrained Swin models were significantly more robust to CT contrast and acquisitions compared to their self-pretrained counterparts. However, the wild-pretraining approach was less beneficial for ViT as well as CNN models. Prior work with natural images(Matsoukas et al., 2022) and medical images(Hosseinzadeh et al., 2021) showed that SSL pretraining is less beneficial for CNN networks. Our analysis with multiple pretext tasks showed higher feature reuse in the lower stages and feature differentiation especially in the later stages for wild-pretrained models. This trend in lower-level feature reuse but differentiation close to higher levels was also observed for CT imaging variations with wild-pretrained models. Further analysis of feature self-similarity showed larger differentiation of features across different layers for wild-pretrained models compared to self-pretrained models, which allows the former models to extract a wider variety of features, which may contribute to higher accuracy and robustness to imaging variations.

## Acknowledgments

This research was partially supported by the NCI R01CA258821 and the Memorial Sloan Kettering Cancer Center Support Grant/Core Grant NCI P30CA008748.

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

## Appendix A. Additional analysis and results

### A.1. Centered kernel alignment (CKA)

Feature similarities between pretrained/self-pretrained and fine tuned models were measured using centered kernel alignment (CKA), which computes a normalized similarity of two feature representations $\boldsymbol{X}$ and $\boldsymbol{Y}$ in terms of the Hilbert-Schmidt Independence Criterion (HSIC):

$$\text{CKA}(\boldsymbol{K}, \boldsymbol{L}) = \frac{\text{HSIC}_0(\boldsymbol{K}, \boldsymbol{L})}{\sqrt{\text{HSIC}_0(\boldsymbol{K}, \boldsymbol{K})\text{HSIC}_0(\boldsymbol{L}, \boldsymbol{L})}} \tag{1}$$

where $\boldsymbol{K}{=}\boldsymbol{X}\boldsymbol{X}^{\boldsymbol{T}}$ and $\boldsymbol{L}{=}\boldsymbol{Y}\boldsymbol{Y}^{\boldsymbol{T}}$ are the Gram matrices of feature $\boldsymbol{X}$ and $\boldsymbol{Y}$. CKA computation typically requires the feature activations of entire dataset to be stored in the memory, which is difficult to implement for transformers that have a large number of parameters. Hence, we implemented the minibatch CKA(Nguyen et al., 2020) by averaging HSIC scores over k minibatches as:

$$\text{CKA}_{\text{minibatch}}(\boldsymbol{K}, \boldsymbol{L}) = \frac{\frac{1}{k}\sum_{i=1}^{k} \text{HSIC}_1(\mathbf{X_i X_i^T}, \mathbf{Y_i Y_i^T})}{\sqrt{\frac{1}{k}\sum_{i=1}^{k} \text{HSIC}_1(\mathbf{X_i X_i^T}, \mathbf{X_i X_i^T})}\sqrt{\frac{1}{k}\sum_{i=1}^{k} \text{HSIC}_1(\mathbf{Y_i Y_i^T}, \mathbf{Y_i Y_i^T})}} \tag{2}$$

An unbiased estimator of HSIC(Song et al., 2012) was computed to reduce dependency of CKA values on the batch size:

$$\text{HSIC}_1(\boldsymbol{K}, \boldsymbol{L}) = \frac{1}{n(n-3)}\left(\text{tr}(\tilde{K}\tilde{L}) + \frac{\mathbf{1}^{\text{T}}\tilde{K}\mathbf{1}\mathbf{1}^{\text{T}}\tilde{L}\mathbf{1}}{(n-1)(n-2)} - \frac{2}{(n-1)}\mathbf{1}^{\text{T}}\tilde{K}\tilde{L}\mathbf{1}\right) \tag{3}$$

### A.2. Additional results

Table A.1. Robustness of tumor segmentation to different scan reconstructions. Significance tests compared wild-pretrained to self-pretrained and scratch trained models using the same network architecture.

| Model | Training | Slice 2.5mm | | | Slice 5mm | | |
|---|---|---|---|---|---|---|---|
| | | Sharp | Smooth | p-value | Sharp | Smooth | p-value |
| CNN | Scratch | 0.20±0.20 | 0.22±0.21 | 0.61 | 0.27±0.24 | 0.28±0.26 | 0.26 |
| CNN | Self-pretraining | 0.21±0.20 | 0.22±0.20 | 0.57 | 0.27±0.19 | 0.31±0.27 | 0.16 |
| CNN | Wild-pretraining | 0.23±0.21 | 0.24±0.23 | 0.68 | 0.30 ± 0.25 | 0.34±0.31 | 0.13 |
| ViT | Scratch | 0.47±0.34 | 0.54±0.32 | 0.08 | 0.49±0.34 | 0.50±0.30 | 0.64 |
| ViT | Self-pretraining | 0.64±0.18 | 0.56±0.25 | 0.019 | 0.58±0.26 | 0.51±0.23 | 0.14 |
| ViT | Wild-pretraining | 0.67±0.16 | 0.58±0.26 | 0.077 | 0.62±0.24 | 0.56±0.22 | 0.11 |
| Swin | Scratch | 0.52±0.32 | 0.36±0.36 | 0.37 | 0.57±0.48 | 0.47±0.34 | 0.12 |
| Swin | Self-pretraining | 0.58±0.27 | 0.54±0.31 | 0.13 | 0.52±0.28 | 0.49±0.30 | 0.058 |
| Swin | Wild-pretraining | 0.70±0.18 | 0.66±0.21 | 0.058 | 0.62±0.28 | 0.58±0.26 | 0.036 |

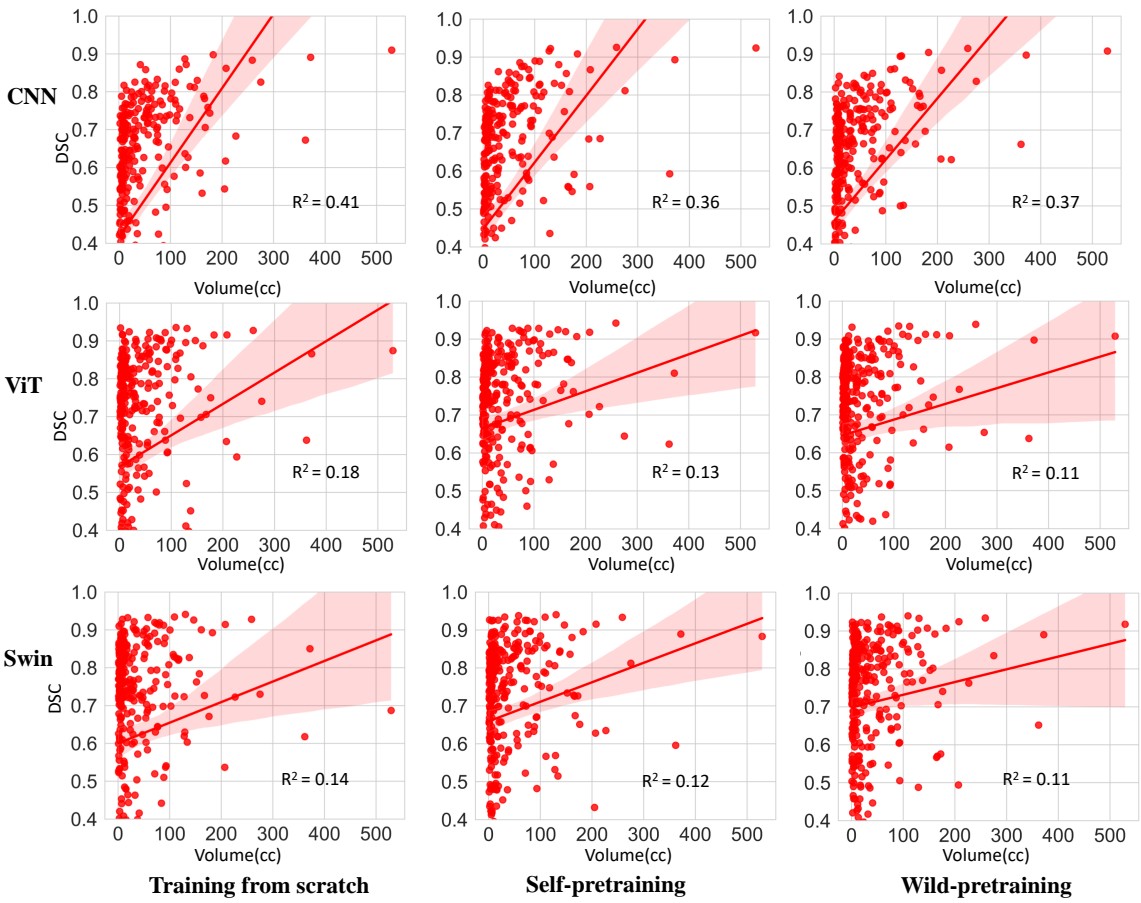

Figure A.1. The scatter plot of DSC versus tumor volume (cc) to assess dependency of accuracy on the tumor volume for the analyzed networks.

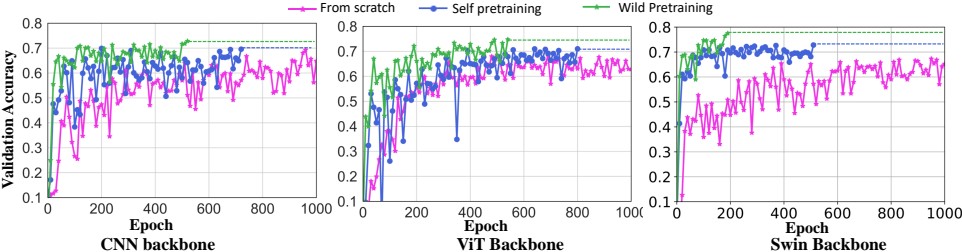

Figure A.2. Finetuning efficiency measured for self-pretrained and pretrained models using CNN, ViT and Swin backbone .

