# OpenReview forum: "Self-supervised pretraining in the wild imparts image acquisition robustness to medical image transformers: an application to lung cancer segmentation"
_MIDL.io/2024/Conference — MIDL 2024 Poster_

### Official Review · Reviewer_wLgp · 2024-02-28

**Confidence:** 4
**Preliminary Rating:** 3
**Final Rating:** 4

**Summary:**

The paper conducts an analysis of pretraining and self-pretraining strategies on transformer models, evaluating their performance on CT segmentation tasks. The experiments demonstrate that pretrained models significantly outperform self-pretrained models for Swin. Additionally, pretrained models require fewer fine-tuning iterations and exhibit greater accuracy and robustness in handling contrast and imaging differences.

**Strengths:**

The motivation is evident, and the evaluation is straightforward to understand.

The paper extensively explores the dataset.

It offers clear implementation details and evaluation metrics, enhancing reproducibility.

**Weaknesses:**

The definition of pretraining and self-pretraining is unclear.

The comparison between ViT and Swin may not be fair due to different number of parameters (46,405,874 V.S. 64,698,114)

The paper lacks CNN baseline and hybrid (CNN-Transformer) baselines.

The paper lacks ablation study. For example, Self-pretraining + pretraining

**Detailed Comments:**

The distinction between pretraining and self-pretraining lacks clarity. Is pretraining a fully supervised method, while self-pretraining a self-supervised method?

The Swin transformer employs a hierarchical structure that could potentially enhance performance compared to ViT. Moreover, the ViT variant chosen in the paper has significantly fewer learnable parameters than Swin; therefore, the comparison may not be equitable.

It remains unclear how many training samples are utilized in self-pretraining (SSL) and how many are used in pretraining. It's commonly understood that SSL typically requires more data to learn generalizable image features.

It would be intriguing to observe an ablation study demonstrating the performance after the model has undergone two-stage pretraining (self-pretraining followed by pretraining).

**Justification Of Final Rating:**

Following a thorough review of the authors' responses and the revisions made to the manuscript, I have revised my evaluation to "weak accept." I anticipate that the improved version will provide enhanced readability and offer deeper insights to the community.

**Justification Of The Preliminary Rating:**

The paper demonstrates a clear motivation, and the dataset selection is comprehensive. However, due to the lack of clarity regarding the training strategy and potential unfair evaluation strategy, the reviewer will assign a borderline rating during the preliminary assessment.

**Questions To Address In The Rebuttal:**

It would be better to receive a response to the comments provided above in the rebuttal.

---

> ### Author Response · Authors · 2024-03-18
> **Rebuttal**
>
> We thank the reviewer for their positive comments and the helpful suggestions. We have revised the manuscript and added experiments as suggested by the reviewer, which we believe has further improved the manuscript.
>
> **Distinction between pretraining and self-pretraining.**
> We have now use “**wild-pretraining**” and “**self-pretraining**” in the paper. We have now clarified the difference between the two approaches in the introduction. As stated in the paper, these are both self-supervised learning (SSL) approaches that differ primarily in the datasets used for pretraining the model. Pretraining makes use of large and uncurated data for the specific task, and diverse datasets for SSL. Self-pretraining is applied on the curated task-specific data. Hence, the number of examples used in the two methods is different, with pretraining often using a much larger number of examples, in our case > 10,000 3D CT volumes for pretraining and 316 3D CT volumes for self-pretraining, respectively.
>
> **Comparison of ViT and Swin**.   As stated in our response to reviewer **CfFt**, our goal was ***not*** to compare Swin and ViT networks as this doesn’t add any new insights than what has been shown in other prior works. We instead focused our statistical comparisons on understanding the differences of using pretraining with uncurated and task-unrelated datasets versus self-pretraining on the curated task-specific dataset applied to the same network architecture. Hence, ViT was compared against itself when using self-pretraining versus pretraining and so on. We have further clarified this in the paper.
>
> **Lack of CNN baseline and hybrid CNN-Transformer.**  We included a CNN-based network called non-skip Unet (nsUnet) developed with SSL techniques (PRCLv2) for medical image segmentation and compared this approach to itself when subjected to pretraining and self-pretraining. Whereas it is interesting to add additional baselines like hybrid CNN-Transformer networks, these require additional and separate pretraining of the individual networks and it’s unclear if they add any additional insights than what is gained by studying the individual network approaches with different inductive biases when using the two different SSL methods. In order to get more insights on the impact of pretraining, we also included additional pretext tasks as suggested by reviewer **8vBC**. Given the page restrictions of the conference paper and to avoid putting too many results in the supplemental section, we chose to keep additional baselines for the journal version of the manuscript.
>
> **Ablation study.**
> We agree and added an analysis that used sequential SSL involving pretraining followed by self-pretraining.

---

> ### Author Response · Authors · 2024-03-19
> **Invitation for discussion**
>
> Dear Reviewer **wLgp**,
>
> Thanks so much for all your comments. We have addressed your comments as best as we can and updated the manuscript accordingly.
>
> Please let us know if you have any questions and we are happy to discuss with you.
>
> Best Regards,

---

### Official Review · Reviewer_8vBC · 2024-02-28

**Confidence:** 4
**Preliminary Rating:** 2
**Final Rating:** 3.5

**Summary:**

The manuscript explores the efficacy of pretraining and self-pretraining strategies in medical image analysis, specifically focusing on the segmentation of lung tumors from 3D-computed Tomography (CT) scans using transformer models (Swin and ViT). The authors compare the two methods in terms of accuracy gains, fine-tuning efficiency, and robustness to variations in image acquisition. The experiments show the superiority of self-pretraining methods to pretraining on different publicly available datasets. This research presents a comparative analysis of pretraining approaches in medical image analysis, providing valuable insights for developing more efficient and adaptable medical imaging models.

**Strengths:**

The paper presents an interesting comparative analysis of pretraining versus self-pretraining in medical image analysis, specifically for lung cancer segmentation from CT scans, using transformer models (Swin and ViT). It demonstrates the superiority of pretraining in terms of accuracy, fine-tuning efficiency, and robustness against imaging variations.

The authors performed an extensive set of experiments across different pre-text tasks and pretraining datasets.

**Weaknesses:**

1) The study's focus on Swin and ViT models may limit the findings' generalizability. Including a broader range of architectures could provide comprehensive insights into pretraining benefits.

2) The results section requires a more comprehensive discussion of the outcomes, particularly regarding the impact of the pretext task and an in-depth investigation into why pretraining outperforms self-training.

3)The manuscript would benefit from improved English language usage, enhancing overall readability and clarity. Moreover, a clearer link between the main paper and the supplementary materials could improve the coherence and utility of the provided resources.

**Detailed Comments:**

Please add a list of contributions in the Introduction Section to clearly outline the paper's achievements and innovations. Moreover, including relevant literature on Transformers would clarify why this architecture was chosen for the analysis.

Expand the discussion to include comparisons with ImageNet-based pretraining strategies, e.g. InceptionV3 and SwaV;

A clearer explanation for interpreting Figure 4 is needed, along with improvements in the visual quality of all figures, including size enhancements and text legibility;

Additional details in Subsection "5.4. Finetuning efficiency" on evaluating fine-tuning efficiency would benefit the research;

Please review the English language used throughout the manuscript. Specific attention should be given to correcting typographical errors (e.g., "fine turning" to "fine-tuning" in Table 1 caption);

Please ensure that spaces are correctly inserted between references and the surrounding text to improve readability.

**Justification Of Final Rating:**

After reviewing the authors' responses and revisions to the manuscript, I have adjusted my evaluation from "weak reject" to "borderline accept." The authors added CNN-based architecture that broadens the study's generalizability beyond the original transformer models (ViT and Swin). The experiment enhances the paper's relevance to more straightforward types of architectures that are yet the most used in medical image analysis. Moreover, the newly added subsection 5.5, which delves into the reasons behind the superior performance of wild-pretraining over self-pretraining, addresses my concern about a "more comprehensive discussion of the outcomes". The authors have improved the English language usage and enhanced figure presentations. These changes address my initial concerns, making the findings more accessible and the paper's contributions clearer.

**Justification Of The Preliminary Rating:**

My preliminary rating of "Weak Reject" is due to the paper's limited scope on Swin and ViT models, which restricts its generalizability. While the study offers novel insights into pretraining strategies in medical image analysis, it needs a comprehensive discussion of the results and the impact of pretext tasks. Additionally, the English quality and figure presentation require significant improvement for clarity.

**Questions To Address In The Rebuttal:**

I recommend the authors demonstrate the generalizability of their findings by including a wider variety of pre-trained networks in their study. Additionally, enhancing the manuscript's clarity through improved English language quality and deepening the discussion on the result.

---

> ### Author Response · Authors · 2024-03-18
> **Rebuttal**
>
> Thank you for the positive comments. We appreciate the suggestions related to “including a broader range of architectures”, “comprehensive discussion”, “impact of pretext task”, etc and have included additional experiments and added more discussion to address these concerns and improved the English to better clarify findings.
>
> **Generalizability of findings.** We now included a CNN based architecture, pretrained using PRCLv2 (published in TPAMI, 2023), which implemented a multi-scale pixel restoration and siamese feature comparison pretext task. We compared its performance when using training from scratch, self-pretraining vs. wild-pretraining on the exact same datasets.
>
> **Comprehensive discussion and pretext tasks.** We added contrastive learning-based pretext task and compared against 4 additional pretext tasks in terms of accuracy on the two testing datasets and studied feature reuse using CKA. We also studied the role of adding self-pretraining on the curated in-domain dataset to the pre-trained model to further understand how these two SSL approaches enhance the performance on downstream tumor segmentation task (two-stage pretraining). We also added an additional CKA analysis to study the similarity of pretrained features in the same versus different layers of the Swin network when it was subjected to pretraining and self-pretraining to understand how the features varied (Figure 5). Our analysis showed larger differentiation of features across different layers with wild-pretraining compared to self-pretraining, which could allow the network to extract different types of features. We added these results in Sec. 5.3, 5.5 and improved our discussion.
>
> ImageNet-based pretraining with SwaV and InceptionV3 are basically 2D based networks and is different from the analysis of 3D CT medical image datasets used in this work. Nonetheless, we added a contrastive learning based pretext task.
>
> **English and clarity.** We apologize for lack of sufficient clarity. We thoroughly revised the paper and added clearer links to the supplementary materials in the main document. We also improved the Figures with larger font sizes for readability (e.g. Figure 4) and improved the explanation. We also revised the introduction and have now included a paragraph on the contributions.
>
>
> PRCLv2: Zhou, Hong-Yu, Chixiang Lu, Chaoqi Chen, Sibei Yang, and Yizhou Yu. "A unified visual information preservation framework for self-supervised pre-training in medical image analysis." IEEE Transactions on Pattern Analysis and Machine Intelligence (2023).

---

> ### Author Response · Authors · 2024-03-19
> **Invitation for discussion**
>
> Dear Reviewer **8vBC**,
>
> Thanks so much for all your comments.
> We have addressed your comments as best as we can and updated the manuscript accordingly.
>
> Please let us know if you have any questions and we are happy to discuss with you.
>
> Best Regards,

---

### Official Review · Reviewer_CfFt · 2024-03-05

**Confidence:** 4
**Preliminary Rating:** 2
**Final Rating:** 3.5

**Summary:**

The manuscript examines the impact of applying self-supervised learning (SSL) to datasets that are distinct from the downstream tasks in comparison to its application on the same downstream dataset. It evaluates two backbone architectures, specifically ViT and Swin Transformer, across two SSL tasks—MIP and ITD—as well as their combined implementations. For the empirical evaluation, the authors employ one public dataset (N=139) and one internal dataset (N=196), both aimed at lung cancer segmentation. The study analyzes several key aspects: segmentation accuracy, fine-tuning efficiency, robustness to data variability, and the feature reusage.

**Strengths:**

+ Overall, the manuscript is well-written and easy to comprehend.
+ The exploration of SSL robustness to data variability (e.g., contrast versus non-contrast, slice thickness, and image reconstruction kernels) is new and important to medical imaging analysis.

**Weaknesses:**

+ Most of the observations and analyses presented in the manuscript have been done in existing studies within both natural and medical imaging fields. The novelty and depth of the methodology and insights offered by this work are limited.
+ Some empirical evaluation conducted with a constrained dataset size raises concerns regarding the robustness of the statistical analysis, potentially undermining the validity of the findings.

**Detailed Comments:**

1. The novelty and depth of the methodology and insights offered by this work are limited. The primary focus of the study is on the influence of curated data on the performance of downstream tasks, a subject previously explored in both natural [1] and medical [2] imaging fields. Furthermore, the observation that Swin generally outperforms ViT is not a novel finding, as reported by Ref [3], which compared ViT and Swin across six different tasks, arriving at a similar conclusion. The aspect of reusing lower-level features has been studied in Ref [4]. Despite this, the manuscript does make a contribution by extending this discussion to the analysis of 3D CT images, but these contributions are somewhat limited and focused solely on the task of lung cancer segmentation.

2. This manuscript offered some interesting analysis regarding the robustness to data variability such as contrast versus non-contrast, slice thickness, and image reconstruction kernels. However, the analysis regarding slice thickness and kernel variation is based on a small sample size of only 20 cases. This limited dataset size may not provide a sufficient basis for drawing meaningful statistical conclusions.


Reference

[1] Goyal, P.,et al. (2021). Self-supervised pretraining of visual features in the wild. arXiv preprint arXiv:2103.01988.

[2] Hosseinzadeh Taher, M. R., et al. (2021). A systematic benchmarking analysis of transfer learning for medical image analysis. In Domain Adaptation and Representation Transfer, and Affordable Healthcare and AI for Resource Diverse Global Health: Third MICCAI Workshop.

[3] Ma, DongAo, et al. "Benchmarking and boosting transformers for medical image classification." MICCAI Workshop on Domain Adaptation and Representation Transfer.

[4] Raghu, Maithra, et al. "Transfusion: Understanding transfer learning for medical imaging." Advances in neural information processing systems 32 (2019).

**Justification Of Final Rating:**

The authors' detailed responses are appreciated. Although concerns regarding the novelty of this work remain, it is acknowledged that most existing research has predominantly focused on 2D rather than 3D, which holds its own practical value. Furthermore, the additional analysis and clarification regarding feature similarity have enhanced this study. Therefore, I have revised my rating to Borderline Accept.

**Justification Of The Preliminary Rating:**

This work exhibits limited novelty and depth, and its observations do not offer substantial insights or generalizability, particularly when considering similar observations in existing studies and the single target task evaluated in this research.

**Questions To Address In The Rebuttal:**

1. The terminology of "Pretraining" and "self-pretraining" may raise confusion since both approaches leverage SSL but on differing datasets. The use of "Pretraining" might be associated with "supervised pretraining," potentially leading to confusion. It would be beneficial to differentiate these terms more distinctly or provide a clear definition to avoid ambiguity.

2. On Page 2, the description of Zhou et al., 2021 is not accurate. In the cited work, the datasets used for pretraining and the downstream tasks differ.

3. The sentence on page 7 regarding self-pretraining is not clear to me: "On the other hand, self-pretraining results in highly similar features with no separation even at higher feature levels for CT contrasts and large variations in low and mid-level features for reconstruction kernels, but not in the higher levels." This statement requires clarification for better understanding.

4. In section 5.3, the observation that datasets pertaining to the task at hand show less feature reuse but higher performance compared to self-pertaining introduces ambiguity regarding the relationship between feature reuse and downstream task accuracy. Clarifying this relationship would enhance understanding. Furthermore, analyzing the performance of pretraining and self-pretraining models used with "off-the-shelf", instead of fine-tuning, may provide valuable insights.

---

> ### Author Response · Authors · 2024-03-18
> **Rebuttal**
>
> Thank you for the positive comments regarding the study of SSL robustness to data variability and also the provided additional references [2] and [3]. We have clarified the novelty and the explanation of results and discussion in response to the reviewer’s suggestions.
>
> ***Our answers to the weaknesses and other stated concerns are below:***
>
> **Limited novelty, influence of curated data on… downstream tasks”** Our goal is to study the impact and the relative benefits of performing SSL wild-pretraining from large, unlabeled, diverse, and uncurated 3D medical image datasets as well as SSL-based unsupervised self-pretraining on curated 3D medical image datasets applied to 3D-based cancer segmentation. Prior works including those suggested by reviewer **CfFt** focused on 2D analysis for detection, classification, normal tissues (lung, blood vessels) and pneumothorax segmentation. We also evaluated how SSL-based wild-pretraining and self-pretraining impacted robustness to CT imaging differences on two independent testing datasets, totalling 335 examples. The prior works [1-4] as provided by reviewer **CfFt** all deal with 2D image analysis and distinctly different problems: [1] dealt with SSL performed for 2D analysis with natural images using transformers, [2] dealt with supervised training with two natural image datasets followed by fine-tuning, SSL performed with natural images followed by SSL on curated medical datasets applied to 2D medical image analysis using ResNet, [3] compared need for SSL with transformers compared to CNNs for 2D medical image classification, [4]  used 2D models supervised pretrained on ImageNet and transfer learned on 2D medical datasets.
>
> **Use of constrained dataset for robustness evaluation.** In total, 335 cases from two independent datasets were tested and used for statistical comparisons (Table 2), 139 from LRad dataset used to study influence of image reconstruction kernels (Figure 2) and contrast differences (Table 3). The 20 examples are additional cases that contained patients reconstructed with two different reconstruction kernels and slice thicknesses and were used to perform paired comparison to study the impact of the SSL pretraining approach on the accuracy by controlling for the tumor and patient anatomy. These results are shown in Supplementary Table A.1.
>
> **Clarification of novelty.**  We ***did*** ***not*** compare the performance of Swin against ViT. Instead, our statistical analyses focused on comparing wild-pretraining and self-pretraining applied to the same network configurations. We have further clarified our explanations. We also added a CNN based method using PRCLv2 as the pretraining pretext.
>
> ***Our response to the concerns in the rebuttal are:***
>
> **1. Clarification between terminology of "Pretraining" and "self-pretraining".** We further clarified the differences in pretraining and self-pretraining. Both of these methods used unsupervised pretraining but with different data instances. Whereas self-pretraining is performed on the curated task-data without exposing any labels using pretext tasks, pretraining applies the same pretext tasks on uncurated, large, and diverse datasets that are distinct from the task-data. To further clarify this, we therefore use the terminology wild-pretraining and self-pretraining in the updated manuscript. Our approach is focused on unsupervised pretraining as used in model genesis. Supervision with labeled data instances is only performed for fine-tuning.
>
> **2.   About reference Zhou et al., 2021.** Thanks for pointing this out. We have now rectified it in Line 7 of introduction.
>
> **3.  Clarification on sentence:** Thank you! We have now clarified in the revised paper。
> “The trend of higher feature reuse for self-pretraining compared to wild-pretrained models was also observed for contrast and non-contrast CT scans (Figure 4). In particular, wild-pretraining resulted in larger deviations of features close to the later encoder layer (13 and 14) for contrast compared to non-contrast CTs (Figure.4  A and B)”
>
> **4. the relationship between feature reuse and downstream task accuracy:**
> Wild-pretraining (Figure 3 E) has higher feature reuse compared to self-pretraining (Figure 3 F) in early and middle layers (1-10). In the deeper layers (especially layer 13 and 14), wild-pretraining tends to have fewer feature reuse compared to self-pretraining, which suggested that wild-pretraining adopted those deeper layers for the segmentation tasks more.
> On the other hand, we also investigate the feature similarity between wild-pretraining and self-pretraining as well as their self-feature similarity, shown in Figure 5. We found that wild-pretrained feature and self-pretrained feature have less similarity (especially after layer 3), shown in Figure (C).  Wild-pretraining features have less similarity compared to self-pretraining, which indicates that wild-pretraining ability to extract a wider range of pretrained feature.

---

> ### Author Response · Authors · 2024-03-19
> **invitation for discussion**
>
> Dear Reviewer **CfFt**,
>
> Thanks so much for all your comments.
> We have addressed your comments as best as we can and updated the manuscript accordingly.
>
> Please let us know if you have any questions and we are happy to discuss with you.
>
> Best Regards,

---

### Author Response · Authors · 2024-03-17

We thank the PC and AC for the opportunity to present our rebuttal and the reviewers for their insightful comments and suggestions. In response to the suggestions, we have made the following changes.

**(1)** further clarified our contributions.

**(2)** improved our explanation of self-pretraining (self supervised learning [SSL] performed on the same curated downstream task dataset without labels) and wild-pretraining (SSL performed on large, diverse, unlabeled and uncurated data that is unrelated to task).

**(3)** further clarified that the performance comparisons were performed between the SSL strategies for the same architecture (e.g. ViT wild-pretrained vs. ViT self-pretrained) instead of comparing ViT versus Swin,

**(4)** added CNN baseline as requested by reviewers.

**(5)** added experiments to study impact of pretext tasks on downstream task accuracy.

**(6)** ablation analysis with two-step pretraining.

**(7)** additional analysis using centered kernel alignment to understand the feature reuse from wild- and self- pretraining to fine-tuning due to the pretext tasks used in SSL as well as to analyze differences in the extracted features across model layers with self-pretraining and wild-pretraining methods in order to improve the discussion of why the accuracies with these approaches differ.

**(8)** deep discussion of the CKA analysis of different pretexts using Swin backbone.

**(9)** further clarified that the evaluation of the robustness imparted from SSL to imaging acquisition variations was applied to 335 test cases and that the additional subset analysis of 20 cases with paired acquisitions available for same patients was done to further understand differences in accuracy with these methods that controlled for the differences due to tumors. We also thoroughly revised the manuscript to rectify typographical errors and improve the English language. All changes made in response to reviewers’ comments are indicated in blue text.

All the revisions are highlighted with blue text. We believe our manuscript is much improved from the prior version and hope the reviewers, AC, and PC would find the manuscript interesting for publication in MIDL.

---

### Meta-Review · Area_Chair_XJjW · 2024-04-04

**Recommendation:** Accept (Poster)
**Confidence:** 4

**Metareview:**

All reviewers agree that the work on SSL proposed here is worthwile and the experiments well-conducted, on a variety of datasets. The reviewers and authors engaged in the rebuttal and scores were improved after rebuttal.

---

### Decision · Program_Chairs · 2024-04-06

Accept (Poster)